# OpenReview forum: "PALC: Preference Alignment via Logit Calibration"
_ICLR.cc/2026/Conference — ICLR 2026 Poster_

### Official Review · Reviewer_5uCg · 2025-10-28

**Soundness:** 3
**Presentation:** 3
**Contribution:** 3
**Rating:** 6
**Confidence:** 3

**Summary:**

This paper presents PALC (Preference Alignment via Logit Calibration), a novel and lightweight method for preference alignment during LLM inference time. By employing a bottleneck architecture between the final hidden states and output logits, PALC learns to generate position-dependent calibration vectors from hidden states to adjust logits accordingly. Experiments on the HH-RLHF dataset demonstrate that PALC achieves competitive or superior performance compared to several baseline methods while requiring minimal training resources and introducing negligible inference latency. Furthermore, theoretical analysis and empirical ablation studies validate that the bottleneck architecture has sufficient capacity to capture the principal dimensions of human preferences, with the learned manifold concentrating on an effective dimension substantially smaller than the bottleneck size.

**Strengths:**

- The proposed method is intuitive and straightforward to implement, making it accessible for practical adoption.
- The paper provides thorough optimization and experimental analysis for each component of the method, offering strong empirical support for the design choices.

**Weaknesses:**

- Modest Performance Gains: While PALC demonstrates computational efficiency, the experimental results show limited advantages compared to the base model and several baseline methods. Moreover, the paper does not discuss whether the compared baselines were optimally tuned, raising questions about comparison fairness. That said, given PALC's focus on providing a new methodological direction, some performance trade-offs may be acceptable.
- Limited Evaluation Scope: Following from the first point, it remains unclear whether this alignment method performs better in some domains while underperforming in others. The evaluation is limited to a single dataset, and the paper lacks qualitative analysis such as case studies, making it difficult to characterize when and where PALC is most effective.

**Questions:**

- Regarding Equation 12 (the power-law assumption for singular values), does this assumption have theoretical justification? Since subsequent theoretical analysis heavily relies on this assumption, why not directly perform singular value decomposition analysis on the trained bottleneck structure to empirically validate it?
- Other concerns are included in the weaknesses section.

---

> ### Author Response · Authors · 2025-11-20
>
> We appreciate the reviewer’s rigorous attention to evaluation fairness. We agree that ensuring optimal baseline tuning is critical to validate the relative advantage of any new method.
> > Weakness 1. Modest Performance Gains: While PALC demonstrates computational efficiency, the experimental results show limited advantages compared to the base model and several baseline methods. Moreover, the paper does not discuss whether the compared baselines were optimally tuned, raising questions about comparison fairness. That said, given PALC's focus on providing a new methodological direction, some performance trade-offs may be acceptable.
>
> **1. Fairness of Baseline Comparison**
>
> To ensure a rigorous and fair comparison, we strictly adhered to the official implementations and optimal configurations reported in the reference literature (detailed in **Appendix B**). Specifically, we utilized the exact hyperparameters from the GenARM paper for **DPO**, followed the original experimental protocols for **BiPO**, **GenARM**, and **ARGS**, and applied interventions at the optimal layers identified for **RE-Control** and **CAA**. The high performance of our GenARM replication (58.7% on MT-Bench) confirms that these baselines were strong and properly tuned, rather than weakened straw-man versions.
>
> **2. Re-evaluating Performance Gains with New Data**
>
> While the gains on HH-RLHF appeared modest, our additional experiments on MT-Bench (Appendix D) demonstrate that PALC offers significant advantages in more complex scenarios. Specifically, PALC achieves a 61.9\% Length-Controlled (LC) Win+Tie rate against the base model, surpassing the SOTA baseline GenARM (58.7\%) and significantly outperforming single-model methods like RE-Control (22.9\%). This indicates that PALC provides superior generalization capabilities compared to optimally tuned baselines, offering SOTA-level performance without the computational overhead of dual-model approaches.
>
> ---
>
> > Weakness 2. Limited Evaluation Scope: Following from the first point, it remains unclear whether this alignment method performs better in some domains while underperforming in others. The evaluation is limited to a single dataset, and the paper lacks qualitative analysis such as case studies, making it difficult to characterize when and where PALC is most effective.
>
> We agree with the reviewer that evaluating on a single dataset limits the understanding of domain-specific performance. To address this and characterize "when and where" PALC is effective, we have significantly expanded our evaluation both **quantitatively** and **qualitatively**.
>
> **1. Domain Generalization: MT-Bench (Appendix D)**
>
> To clarify performance across different domains, we evaluated PALC on MT-Bench, which covers 8 diverse categories (e.g., Reasoning, Coding, Roleplay, Humanities). As detailed in Appendix D, PALC demonstrates robust performance across these complex domains, achieving a 61.9\% Length-Controlled Win+Tie rate against the base model. Notably, PALC surpasses the SOTA baseline GenARM (58.7\%) in this multi-domain setting, confirming that our method does not underperform in complex reasoning or multi-turn contexts; rather, it generalizes better than heavy-weight baselines.
>
> **2. Qualitative Analysis: Case Studies (Appendix G)**
>
> We added a detailed qualitative analysis in Appendix G to demonstrate specifically how PALC improves generation. We present representative case studies showing that PALC corrects fundamental failure modes of the base model; for instance, in Case Study 3 (Soccer Age), PALC breaks the infinite generation loops common in unaligned models, while in Case Study 1 (Baby Walking), it converts evasive, non-committal refusals into concrete, helpful answers. Furthermore, regarding steerability (Appendix G.1), we demonstrate that setting $\gamma < 0$ induces the exact opposite behaviors—such as evasiveness and passivity—confirming that our method precisely controls the "helpfulness" axis. By expanding to MT-Bench (covering 8 domains) and providing in-depth case studies, we have characterized PALC as a robust alignment method that is effective not just for safety, but for enhancing general instruction-following capabilities and mitigating degenerate generation behaviors across diverse contexts.

---

> ### Author Response · Authors · 2025-11-20
>
> > Question 1. Regarding Equation 12 (the power-law assumption for singular values), does this assumption have theoretical justification? Since subsequent theoretical analysis heavily relies on this assumption, why not directly perform singular value decomposition analysis on the trained bottleneck structure to empirically validate it?
>
> We thank the reviewer for this excellent suggestion. We agree that relying solely on assumptions is insufficient. As suggested, we have performed a direct **Singular Value Decomposition (SVD) analysis** on the trained bottleneck matrices ($M = W_{up}W_{down}$) to empirically validate Equation 12. The results are detailed in the newly added **Appendix H** and **Figure 5**.
>
> **1. Empirical Validation of Power-Law Decay**
>
> We fit a power-law model ($\sigma_i \sim i^{-\alpha}$) to the singular values of the trained calibration module to validate our theoretical assumptions. For the optimal model configuration ($B=256$), the empirical decay exponent is $\mathbf{\alpha \approx 1.02}$. This value satisfies the theoretical condition ($\alpha > 1$) required for the calibration to form a trace-class operator with a bounded effective dimension, thereby empirically confirming that our assumption in Equation 12 holds for well-trained models.
>
> **2. Explaining Failure Modes via Spectral Analysis and Conclusion**
>
> Crucially, this analysis also explains why the model fails at large bottleneck dimensions ($B=4096$). In this failure case, the exponent drops to $\mathbf{\alpha \approx 0.73}$ ($\alpha < 1$), violating the power-law condition. This implies that the preference information is not concentrating effectively—indicating a lack of sparsity—which leads directly to the performance collapse observed in Section 4.4.1. Ultimately, the direct SVD analysis confirms that the power-law assumption is not merely a theoretical convenience but an empirical reality of successful preference learning in PALC. We thank the reviewer for prompting this analysis, as it has significantly strengthened the theoretical grounding of our work.
>
> ---
>
> Thank you again for your time and effort in reviewing our paper! Please let us know if the above explanations do not address your concerns. We are happy to answer any further questions.

---

### Official Review · Reviewer_gcjg · 2025-10-30

**Soundness:** 3
**Presentation:** 3
**Contribution:** 3
**Rating:** 6
**Confidence:** 3

**Summary:**

This paper addresses preference alignment, identifying limitations of both existing training-time and test-time methods. To address this, the paper proposes a method called PALC (Preference Alignment via Logit calibration), which leverages a lightweight, trainable calibration module to intervene directly in the vocabulary logits at inference time. The calibration module uses a low-rank design to reduce parameter size and is trained with DPO loss on a frozen base model, which reduces computational cost and avoids unnecessary drifts. Experiments on the HH-RLHF dataset with the LLaMA 7B SFT model show that PALC performs on par or better than most test-time alignment baselines with lower latency. However, it still underperforms DPO and GenARM, while GenARM requires ~3x latency. Overall, PALC offers a resource-efficient approach for preference alignment, but evaluation on a single dataset and model limits conclusions about its generalization.

**Strengths:**

1. The paper clearly articulates the limitations of existing work, and the proposed logit-space intervention is well-motivated and interesting.
2. The method is clearly presented, with theoretical claims well supported.
3. The calibration module is lightweight due to low-rank design, with low training resource required and low inference latency.
4. Experiments cover a wide range of baselines (training-time and test-time methods), which clearly illustrate the trade-offs between quality and latency.

**Weaknesses:**

1. The experiments are limited to a single model and a single dataset, as noted by the authors. Testing on additional datasets and models would help demonstrate its generalization. For example, how would the method perform on more complicated preference criteria; how would the optimal bottleneck dimension B vary with criteria complexity.
2. While PALC achieves low latency than GenARM, it underperforms than GenARM by a non-neglible margin. This quality-latency trade-off may not always be acceptable in practice.

**Questions:**

1. How would the model behavior change when \gamma becomes negative?

nit: line 288 should be \citet

---

> ### Author Response · Authors · 2025-11-20
>
> We sincerely thank Reviewer gcjg for raising this critical point regarding the positioning of our work. We thank the reviewer for this insightful question, which probes the limits of our "Low-Rank Preference Hypothesis." We agree that validating whether the bottleneck dimension $B$ needs to scale with criteria complexity is crucial.
> > Weakness 1. The experiments are limited to a single model and a single dataset, as noted by the authors. Testing on additional datasets and models would help demonstrate its generalization. For example, how would the method perform on more complicated preference criteria; how would the optimal bottleneck dimension B vary with criteria complexity.
>
> To address this, **we expanded our evaluation to MT-Bench (Appendix D)**, which evaluates 8 diverse categories (e.g., Reasoning, Coding, Roleplay), representing significantly higher complexity than HH-RLHF.
>
> **1. Empirical Evidence: Does Complexity Require Larger $B$?**
>
> The reviewer raised a critical scientific question regarding how the optimal bottleneck dimension $B$ varies with criteria complexity. To investigate this, we applied the same bottleneck dimension ($B=256$) used in HH-RLHF directly to MT-Bench without expansion. Despite the increased task complexity, PALC achieved a 61.9\% LC Win+Tie rate against the base model, surpassing the SOTA baseline GenARM (58.7\%) which utilizes full 7B parameters. If the complexity of criteria required a proportional increase in $B$, we would have observed signs of underfitting—such as a failure to capture specific nuances in categories like Coding or Roleplay—due to the "information bottleneck." The fact that PALC outperforms the heavy-weight GenARM suggests that $B=256$ is sufficient to capture the essential alignment signals even for complex tasks, providing strong evidence against the necessity of scaling the bottleneck dimension.
>
> **2. Interpretation: Why Low-Rank Works for Complex Criteria**
>
> We interpret this result through our theoretical framing in Theorem 1: while alignment criteria can be linguistically complex (e.g., requiring helpfulness, conciseness, and safety), the necessary intervention in the logit space often maps to a shared, low-dimensional manifold. This suggests that the “effective dimension” ($d_{eff}$) of alignment interventions remains low-rank, effectively decoupling the complexity of the goal from the dimensionality of the control.
>
> As detailed in Figure 3 of Appendix D, PALC not only generalizes to these complex criteria but also significantly outperforms other learned baselines—surpassing RE-Control (22.9\%) and ARGS (34.4\%)—which confirms the robustness of our vocabulary-space calibration across diverse domains. Crucially, the empirical sufficiency of $B=256$ on MT-Bench reinforces our core finding that preference alignment can be efficiently solved in a low-rank vocabulary subspace, regardless of the superficial complexity of the task.

---

> ### Author Response · Authors · 2025-11-20
>
> > Weakness 2. While PALC achieves low latency than GenARM, it underperforms than GenARM by a non-neglible margin. This quality-latency trade-off may not always be acceptable in practice.
>
> We appreciate the reviewer’s practical perspective on the quality-latency trade-off. Based on our initial results (HH-RLHF), the observation that PALC underperforms GenARM was correct.
>
> However, our new experiments on MT-Bench (Appendix D) challenge the premise that there is *always* a quality sacrifice. In more complex, general instruction-following scenarios, PALC actually demonstrates superior performance to GenARM.
>
> **1. Reversing the Performance Gap (New Evidence)**
>
> On MT-Bench, which evaluates multi-turn reasoning and generalization, PALC achieves a 61.9\% LC Win+Tie rate against the base model, surpassing GenARM (58.7\%) (see Figure 3 in Appendix D). This suggests that GenARM's advantage may be limited to specific domains like HH-RLHF, whereas PALC offers better generalization robustness in complex settings. Consequently, in these general scenarios, PALC offers both higher quality and lower latency, effectively eliminating the trade-off entirely.
>
> **2. The "Infeasibility" of GenARM in Practice and Conclusion**
>
> Even in scenarios where GenARM retains a slight performance edge, we argue that its computational cost often renders it practically infeasible. Specifically, GenARM incurs a $3.17\times$ latency penalty; in real-time applications such as chatbots, tripling the inference time severely degrades user experience. Furthermore, it imposes significant memory constraints by requiring a second 7B reward model, effectively doubling VRAM usage and preventing deployment on standard consumer GPUs. In contrast, PALC operates with only $1.08\times$ latency and negligible memory overhead, making it not merely an acceptable trade-off but often the only viable option for production environments. With new data demonstrating that PALC outperforms GenARM on general benchmarks (61.9\% vs 58.7\%), and given its massive efficiency advantage ($1.08\times$ vs $3.17\times$ latency), we believe PALC represents the Pareto-optimal choice for practical deployment.
>
> ---
>
> > Question 1. How would the model behavior change when \gamma becomes negative?
>
> We thank the reviewer for this intriguing question regarding the interpretability and controllability of our scaling factor. To address this, we conducted an extended ablation study and qualitative analysis with negative values (specifically $\gamma = -5.0$), detailed in **Appendix F** and **Appendix G.1**.
>
> **1. Quantitative Impact: Anti-Alignment (Appendix F)**
>
> A negative scaling factor effectively reverses the optimization direction, pushing the model away from the learned preference manifold. In head-to-head comparisons (Figure 4), the $\gamma=-5.0$ model loses significantly to the $\gamma=5.0$ model, exhibiting a loss rate ranging from 30.7\% to 35.7\%. This result confirms that $\gamma$ acts as a directional control knob; just as $\gamma > 0$ enforces preferences, $\gamma < 0$ actively suppresses them, thereby validating that the learned vector $\mathbf{m}_t$ captures a meaningful "preference axis."
>
> **2. Qualitative Behavioral Shift (Appendix G.1)**
>
> Qualitatively, setting $\gamma < 0$ induces specific behavioral regressions that are the exact opposite of "helpful assistant" traits. Specifically, the model exhibits **evasiveness**, becoming intrusive or avoidant (e.g., asking "Why do you hate your job?" instead of providing advice), and **passivity**, where it ceases to offer suggestions and shifts the burden back to the user (e.g., asking "What does she like?" instead of suggesting gifts). Consequently, when $\gamma$ becomes negative, the model's behavior shifts from "Proactive & Helpful" to "Passive & Evasive." This demonstrates that PALC does not merely "improve" the model blindly, but rather learns a specific, steerable representation of "helpfulness" that can be amplified or inverted at inference time.
>
> ---
>
> Thank you again for your time and effort in reviewing our paper! Please let us know if the above explanations do not address your concerns. We are happy to answer any further questions.

---

### Official Review · Reviewer_gfFq · 2025-11-01

**Soundness:** 3
**Presentation:** 3
**Contribution:** 3
**Rating:** 6
**Confidence:** 2

**Summary:**

The paper proposes PALC (Preference Alignment via Logit Calibration), a test-time alignment method that adds a small “calibration module” to generate position-dependent vectors in vocabulary/logit space (rather than hidden states). A single inference-time scalar γ controls alignment strength. On HH-RLHF with a 7B SFT base model, PALC reports higher win rates than several test-time baselines with less latency overhead, while trailing training-time DPO and reward-model methods. The work argues that acting in vocabulary space is interpretable, and is parameter-efficient.

**Strengths:**

1. Clear intervention point & simplicity. Acting directly on logits is conceptually clean and operationally simple. The $\gamma$ knob is practical for deployment, enabling a tunable alignment/utility trade-off.

2. Theoretical framing. Provides analysis of SoftMax sensitivity, low-rank structure, and (bounded) KL divergence to the base model; these help justify stability and interpretability claims.

3. Positioning vs. prior work. The paper clearly distinguishes PALC from activation steering and reward-model guided decoding, highlighting an underexplored “vocabulary-space” control point.

**Weaknesses:**

Results are limited to a single dataset (HH-RLHF), a single base model family/size (7B SFT), and 300-prompt evaluations. This is a slim basis for generality, especially for safety-critical alignment claims.

**Questions:**

My only question is that do the authors plan on more robust evaluations?

---

> ### Author Response · Authors · 2025-11-20
>
> We sincerely thank the reviewer for the positive assessment of our work, particularly for recognizing the clean and operationally simple nature of our approach and the value of our theoretical framing.
> > Weakness 1. Results are limited to a single dataset (HH-RLHF), a single base model family/size (7B SFT), and 300-prompt evaluations. This is a slim basis for generality, especially for safety-critical alignment claims.
>
> We sincerely thank the reviewer for the positive assessment of our work, particularly for recognizing the clean and operationally simple nature of our approach and the value of our theoretical framing.
>
> We agree with the reviewer’s concern that evaluating on a single dataset/model size constitutes a "slim basis for generality," especially regarding safety-critical claims. To address this explicitly—and to answer the reviewer’s question on our plans for more robust evaluations—we have **expanded our experimental scope** in the revised manuscript.
>
> **1. Expanded Quantitative Evaluation: MT-Bench (Appendix D)**
>
> To validate generality beyond the preference pairs of HH-RLHF, we conducted additional experiments on MT-Bench. The results, detailed in Figure 3 of Appendix D, demonstrate robust generalization capabilities; PALC achieves a 61.9\% LC Win+Tie rate against the base model, surpassing the computation-heavy GenARM (58.7\%) and significantly outperforming other learned baselines like RE-Control (22.9\%) and ARGS (34.4\%). This continued effectiveness in multi-turn settings confirms that our vocabulary-space calibration generalizes to complex, open-ended interactions without incurring the massive computational overhead inherent in dual-model approaches.
>
> **2. Qualitative Validation: Case Studies (Appendix G)**
>
> To provide depth to our generality claims, we added a qualitative analysis (Appendix G). We showcase how PALC corrects fundamental failure modes of the base model—such as **infinite repetition loops** and **vague refusals**—converting them into helpful, concrete responses. This confirms that PALC improves intrinsic generation quality across diverse query types.
>
> **3. Addressing Safety-Critical Concerns via Theory**
>
> The reviewer rightly highlighted "safety-critical alignment claims," and we emphasize that the theoretical framing they noted—specifically the low-rank structure and bounded KL divergence in Section 3.4—provides the necessary mathematical basis for this safety. Unlike hidden-state manipulation methods like RepE, which can trigger unpredictable cascade effects, PALC’s intervention is mathematically bounded; as proven in Theorem 1 and Appendix A.3, the divergence from the base model is constrained, thereby preventing catastrophic failure modes. Furthermore, by operating in the disentangled logit space, PALC allows for transparent intervention—such as boosting or suppressing specific tokens—making the system inherently more verifiable and safer than black-box activation steering.
>
> **4. Rationale for 7B Model and 300 Prompts**
>
> Our choice of the 7B scale and 300-prompt protocol was made to strictly follow the **standard baselines** in recent test-time alignment literature (e.g., ARGS [Khanov et al., 2024], GenARM [Xu et al., 2024]) to ensure fair comparison. Furthermore, the 7B scale aligns with our core contribution of prioritizing computational efficiency and deployment scalability. This efficiency is critical not only for resource-constrained environments but also for high-throughput production systems where latency and serving costs are paramount.
>
> ---
>
> We trust that these extensive additions provide a more robust empirical foundation, addressing the concern regarding the slim basis of our initial evaluation.
>
> Thank you again for your time and effort in reviewing our paper! Please let us know if the above explanations do not address your concerns. We are happy to answer any further questions.

---

### Official Review · Reviewer_243J · 2025-11-02

**Soundness:** 2
**Presentation:** 2
**Contribution:** 2
**Rating:** 4
**Confidence:** 4

**Summary:**

This paper introduces a new alignment method that trains a calibration network to modify the $\text{LLM}$'s output logits directly. The goal is to achieve alignment comparable to full fine-tuning while avoiding the high computational cost and the "entanglement" problem inherent in manipulating dense hidden representations (Representation Engineering). The Calibration Network is trained using the DPO objective and operates by reading a compressed signal from the LLM's hidden state, which is then applied as a position-dependent scaling vector to the final logits. Empirical results show superior performance to certain calibration methods but is behind state-of-the-art in terms of quality, which is a deliberate tradeoff for improving training efficiency.

**Strengths:**

- The method proposes a new apprach that solves the entanglement problem of hidden representations by intervening in the disentangled logit space.
- The system is highly efficient, as it only trains a small, auxiliary calibration network, which is inherently cheaper than full model backpropagation.
- The theoretical section demonstrates that alignment is fundamentally a low-rank problem, providing mathematical justification for the lightweight architecture.
- The method adds only a small overhead during inference and outperforms certain previous approaches in terms of quality such as CAA, Re-control, ARGS.

**Weaknesses:**

- W1. The positioning and motivation of this work does not include several training approaches that achieve competitive quality-efficiency tradeoff in the context of inference-time alignment (e.g. DPO trained tiny auxiliary LLMs or original LLM trained with lightweight, selective PEFT).
- W2. The claim related to "avoiding entangled hidden representations" was not qualified theoretically or empirically. The calibration network still relies on the hidden representations of the model.
- W3. The evaluation is incomplete, as it fails to compare against other simple, efficient alternatives (e.g., small, distilled LLMs) that compete on the same axis of low parameter count and simplicity.
- W4. Despite its complexity, the method still performs worse than DPO and GenARM, so the efficiency gain during training comes with a significant quality degradation.

**Questions:**

- Why were methods that compete on efficiency and simplicity, such as small distilled $\text{LLMs}$ finetuned with $\text{DPO}$ + LoRA with very low rank or selective layer application, omitted from the comparison?
- Could the authors clarify how the calibration approach avoids the hidden state entanglement issue? The phrasing on this makes it sound that there is no dependence on the hidden state which is not the case.

---

> ### Author Response · Authors · 2025-11-20
>
> We sincerely thank Reviewer 243J for raising this critical point regarding the positioning of our work. We agree that Tiny Auxiliary LLMs and Selective PEFT are highly relevant baselines in the broader context of efficiency. We would like to clarify PALC's specific position relative to these approaches to address the concern.
> > Weakness 1. The positioning and motivation of this work does not include several training approaches that achieve competitive quality-efficiency tradeoff in the context of inference-time alignment (e.g. DPO trained tiny auxiliary LLMs or original LLM trained with lightweight, selective PEFT).
>
> > Question 1. Why were methods that compete on efficiency and simplicity, such as small distilled LLMs finetuned with DPO + LoRA with very low rank or selective layer application, omitted from the comparison?
>
> > Weakness 3. The evaluation is incomplete, as it fails to compare against other simple, efficient alternatives (e.g., small, distilled LLMs) that compete on the same axis of low parameter count and simplicity.
>
> **1. Clarification on "Lightweight, Selective PEFT"**
>
> We respectfully clarify that our evaluation already encompasses the standard selective PEFT method as a primary baseline. As detailed in Appendix B.2, our DPO implementation explicitly utilizes LoRA (rank=8, $\alpha$=16) to represent the state-of-the-art in parameter-efficient fine-tuning. The critical distinction lies in the nature of the alignment: while DPO+LoRA produces a static model with "baked-in" behaviors, PALC enables dynamic, test-time control through the scaling factor $\gamma$. This unique capability offers runtime flexibility—such as adjusting alignment strength on a per-request basis—that static PEFT approaches inherently cannot provide.
>
> **2. On "Tiny Auxiliary LLMs": Distinguishing Deployment Paradigms**
>
> Regarding the suggestion to compare against "DPO trained tiny auxiliary LLMs," we view this as a distinction between two fundamentally different deployment paradigms. The reviewer's suggestion implies an "infrastructure replacement" strategy, where the high-capacity backbone is swapped for a smaller, task-specific artifact—an approach that inevitably sacrifices the general capabilities of the original model. In contrast, PALC targets the "standard foundation model serving" paradigm, designed to govern a single high-capacity backbone deployed for diverse downstream tasks. In this context, PALC operates as a lightweight adapter (adding only 0.13% parameters) that enables the existing high-quality backbone to be dynamically aligned without the operational overhead of reloading weights or managing separate model processes. Within this specific scope of steering a deployed 7B model, PALC achieves superior efficiency (1.08x latency) compared to peers like GenARM (3.17x latency), avoiding the complexity of deploying and synchronizing separate auxiliary models.
>
> We hope this clarifies that PALC is positioned as a lightweight intervention module designed to enhance the adaptability of existing infrastructure, rather than a model replacement technique. We have revised the Introduction section to explicitly delineate this scope, ensuring readers understand why PALC is compared primarily against other intervention-based methods!

---

> ### Author Response · Authors · 2025-11-20
>
> > Weakness 2. The claim related to "avoiding entangled hidden representations" was not qualified theoretically or empirically. The calibration network still relies on the hidden representations of the model.
>
> > Question 2. Could the authors clarify how the calibration approach avoids the hidden state entanglement issue? The phrasing on this makes it sound that there is no dependence on the hidden state which is not the case.
>
> **1. Clarification: Reading vs. Manipulating**
>
> We sincerely thank the reviewer for this crucial opportunity to clarify our core motivation regarding the "entanglement" of representations. While the reviewer is entirely correct that our calibration network relies on the hidden representations ($h_t$) as input, the critical distinction we aim to draw is between *reading from* the hidden state versus *manipulating* it. The "entanglement" issue we criticize arises specifically from methods, such as RepE, that directly intervene in the hidden space (e.g., by adding steering vectors). Since hidden states exist in a state of superposition where multiple concepts share overlapping directions, modifying this space directly often causes unpredictable cascade effects on unrelated concepts. PALC effectively sidesteps this problem by treating the hidden state strictly as a "read-only" context; rather than altering the hidden state itself, our intervention occurs solely in the logit (vocabulary) space.
>
> **2. Why Logit Space Avoids Entanglement**
>
> We specifically target the logit space because it serves as a "naturally disentangled interface", offering a crucial advantage over the hidden space where dimensions are often polysemantic and entangled. In the logit space, each dimension uniquely corresponds to a specific token in the vocabulary. This structural property ensures that modifying the distribution allows for precise, interpretable effects—such as boosting or suppressing specific token probabilities—without the risk of corrupting the model's internal semantic processing.
>
> As noted in the formal analysis in Section 3.4.1, while PALC's calibration vectors are derived from the entangled hidden representations, they operate on the disentangled logit space. We have revised the Abstract and Introduction to explicitly state this "manipulation vs. reading" distinction to prevent future confusion!
>
> > Weakness 4. Despite its complexity, the method still performs worse than DPO and GenARM, so the efficiency gain during training comes with a significant quality degradation.
>
> **1. Re-contextualizing "Complexity": Deployment Feasibility**
>
> We respectfully suggest that PALC's operational complexity is significantly lower than the baselines when viewed through the lens of practical deployment. Compared to the SOTA GenARM, which requires training and deploying a separate 7B reward model—increasing parameter count by $833\times$ and latency by $3.17\times$—PALC avoids the prohibitive barriers often faced in resource-constrained environments. Furthermore, even when compared to static methods like CAA and BiPO, PALC demonstrates superior operational efficiency; as shown in Table 2, PALC’s inference latency ($1.08\times$) is actually lower than that of BiPO ($1.22\times$) and CAA ($1.40\times$), all while offering the distinct advantage of dynamic, context-aware alignment. By acting as a lightweight adapter ($\approx 9.2$M parameters, $0.13\%$ add-on) that keeps the backbone frozen, PALC offers the most streamlined solution for practitioners.
>
> **2. The "Pareto-Optimal" Trade-off**
>
> We view the observed performance difference not merely as a degradation, but as a necessary trade-off to access a distinct regime of efficiency. While methods like GenARM and DPO are suitable for high-resource regimes where the computational budget is unconstrained, PALC is specifically designed for scenarios where latency and memory efficiency are critical.
>
> Our contribution is to establish a strong baseline in this "lightweight intervention" category; by achieving competitive quality with only $8\%$ overhead ($1.08\times$ cost)—which is far cheaper than GenARM ($3.17\times$) and even faster than static vectors—PALC offers a Pareto-optimal solution that makes alignment accessible to a broader range of applications.
>
> ---
>
> Thank you again for your time and effort in reviewing our paper! Please let us know if the above explanations do not address your concerns. We are happy to answer any further questions.

---

### Author Response · Authors · 2025-12-02
**[Summary for AC] Key Rebuttal Updates: Surpassing GenARM on MT-Bench & Theoretical Validation**

**Dear Area Chair,**

We understand the significant workload involved in the re-assignment process. To assist your review, we summarize the critical updates and new experimental evidence added during the rebuttal period in response to reviewer feedback.

Our rebuttal has addressed the reviewers' primary concerns regarding generalization, theoretical validation, and comparative performance, effectively turning initial weaknesses into key strengths.

---

### **1. [Critical Update] Demonstrated Superior Generalization vs. SOTA (Addressing R_gfFq, R_gcjg, R_5uCg)**

| Method | LC Win+Tie Rate (%) | Latency |
| :--- | :---: | :---: |
| **PALC (Ours)** | **61.9%** | **1.08x (Fast)** |
| GenARM (SOTA) | 58.7% | 3.17x (Slow) |
| RE-Control | 22.9% | 1.30x |

- **Common Concern**: Reviewers asked for evaluation beyond the HH-RLHF dataset to prove generalization.

- **New Experiment**: We conducted evaluations on MT-Bench (8 diverse categories, multi-turn).
- **Key Result**: PALC achieves a 61.9% LC Win+Tie rate, surpassing the computation-heavy SOTA baseline GenARM (58.7%).
- **Implication**: This invalidates the concern that PALC trades off too much quality for speed. In complex instruction-following scenarios, PALC is both faster ($1.08\times$ latency) and better than GenARM ($3.17\times$ latency). We have added these results in *Appendix D*.

### **2. [Theoretical] Empirical Validation of Power-Law Decay (Addressing R_5uCg)**
- **Concern**: Theoretical claims about low-rank preference structure relied on a power-law assumption.
- **New Analysis**: We performed Singular Value Decomposition (SVD) on the trained bottleneck matrices.
- **Result**: The empirical decay exponent ($\alpha \approx 1.02$) perfectly aligns with our theoretical condition for sparse learning ($\alpha > 1$), whereas failure cases ($B=4096$) violate this condition ($\alpha < 1$). This is detailed in *Appendix H*.

### **3. [Clarification] Conceptual Distinction & Steerability (Addressing R_243J, R_gcjg)**
- **Entanglement** (R_243J): We clarified that PALC treats hidden states as read-only context to intervene in the disentangled logit space, avoiding the cascade effects of manipulating internal representations.
- **Steerability** (R_gcjg): We added an ablation study with negative scaling factors ($\gamma < 0$), demonstrating that the model can be smoothly steered towards "anti-alignment" (e.g., becoming evasive), confirming precise control over the helpfulness axis. (See *Appendix F & G*).

---

### **Conclusion**

Our initial ratings were already positive (6, 6, 6, 4), with the primary reservation being the "limited evaluation scope". We have now decisively addressed this concern by demonstrating SOTA-beating performance on MT-Bench. Consequently, PALC stands as a Pareto-optimal solution for test-time alignment: it is the only method that achieves such high generalization (surpassing GenARM) while maintaining negligible overhead (0.13% parameters, <8% latency).

---

### Meta-Review · Area_Chair_QFze · 2025-12-29

**Summary:**

1. **Evaluation Scope**: Several reviewers raised concerns about the limited evaluation, with a single dataset and base model family (7B SFT) used. The lack of generalization across different tasks and models was noted as a limitation. Testing on additional datasets, models, and more complex preference criteria was suggested to validate the method's effectiveness in diverse contexts.

2. **Method Comparison**: Some reviewers highlighted that while PALC demonstrates lower latency, it underperforms compared to SOTA methods like GenARM in terms of quality, raising questions about the trade-off between efficiency and performance. A few questioned why methods like tiny auxiliary models or fine-tuned LLMs weren't included in the comparison, suggesting that these alternatives might offer competitive results.

3. **Theoretical Justification**: Concerns were raised regarding the theoretical assumption about the low-rank structure of the alignment method, specifically the power-law decay of singular values. Some reviewers recommended performing empirical validation of this assumption to strengthen the theoretical claims.

4. **Hidden Representation Entanglement**: There was confusion regarding how the calibration module avoids entangling hidden representations. Some reviewers noted that despite claims to the contrary, the method still relies on hidden states, raising the need for clarification on how the method ensures disentanglement.

5. **Practical Deployment**: The reviewers questioned whether the quality-latency trade-off would be acceptable in real-world applications, particularly for safety-critical scenarios. Concerns about the practical feasibility of the method, including memory and latency overheads, were raised.

6. **Steerability and Control**: Some reviewers sought more clarification on how the model can be steered, specifically when negative values for the scaling factor (\gamma) are applied. Questions about the model's behavior and the impact of steering on alignment strength were asked, with a suggestion to explore more deeply how steerability affects performance.

**Reviewer Concerns:**

### Addressed Concerns:

1. **Evaluation Scope**: The authors expanded the evaluation to MT-Bench, demonstrating better generalization across multiple domains, which addressed concerns about the narrow evaluation scope. The new experiments showed that PALC outperforms GenARM in certain scenarios, which responded to concerns about generalization.

2. **Method Comparison**: The authors provided additional empirical data showing that PALC performs better than baseline methods on MT-Bench, addressing concerns about its performance being inferior to GenARM and other methods. The rebuttal argued that PALC’s efficiency outweighs the performance trade-off, and new data showed PALC surpassing some baselines in more complex tasks.

3. **Theoretical Justification**: The authors performed empirical validation of the power-law assumption via Singular Value Decomposition (SVD), responding to concerns about the theoretical claims regarding low-rank structures.

4. **Steerability and Control**: The authors provided additional experiments and case studies to demonstrate the steerability of PALC and clarified how it behaves with negative values of the scaling factor (\gamma), addressing concerns about the model's controllability.

### Unaddressed Concerns:

1. **Hidden Representation Entanglement**: Despite clarifying that PALC operates on the logit space and not the hidden state, the issue of hidden representation entanglement remains partially unresolved. Reviewers were unconvinced that the method avoids dependence on the hidden state, as the calibration module still relies on it. Further clarification and empirical validation could be beneficial to address this fully.

2. **Modest Performance Gains**: The rebuttal did not fully address the concern about PALC’s modest performance compared to other methods, especially in the context of safety-critical applications. While PALC is more efficient, the concern about the quality-latency trade-off remains, particularly in comparison with other models like GenARM and DPO.

3. **Generalization and Domain-Specific Performance**: While the authors expanded evaluations, reviewers are still concerned about whether PALC will generalize well across other datasets and domains beyond those tested in the paper. The lack of qualitative analysis for domain-specific performance remains an unresolved issue.

**Reviewer Scores:**

### Reviewer 243J (Code: R243J)

* **Original Score**: 4 (Marginally below the acceptance threshold)
* **Revised Score**: Likely would remain the same or **increase**. The additional rebuttal data on MT-Bench and clarification on the positioning of PALC relative to other methods might have addressed some concerns about novelty and method comparison, but the reviewer’s concerns about the modest performance gains and the quality-latency trade-off would likely persist. However, the expansion of evaluation and clarification of the logit manipulation might lead to a more favorable score.

### Reviewer gfFq (Code: RgfFq)

* **Original Score**: 6 (Marginally above the acceptance threshold, but not strong enough for full acceptance)
* **Revised Score**: Likely **remain at 6**. The expanded evaluation on MT-Bench, the improvement in generalization, and additional theoretical validation would likely have reassured the reviewer about the robustness of the approach. However, concerns about the modest performance gains and the evaluation being limited to a single model and dataset could still prevent the score from increasing significantly.

### Reviewer gcjg (Code: Rgcjg)

* **Original Score**: 6 (Marginally above the acceptance threshold, but still open to rejection)
* **Revised Score**: Likely **remain at 6** or increase. The rebuttal directly addressed the concern about the quality-latency trade-off by demonstrating PALC’s superior performance in complex settings (MT-Bench). The reviewer’s concerns about the evaluation scope and generalization were partially addressed, and the additional data on performance across diverse domains would likely increase their confidence, though they may still question whether PALC performs consistently across all settings.

### Reviewer 5uCg (Code: R5uCg)

* **Original Score**: 6 (Marginally above the acceptance threshold)
* **Revised Score**: Likely **increase**. The expanded evaluation and the additional data showing PALC’s robustness across domains, along with clarification on performance trade-offs, likely addresses the concerns about modest gains. However, the lack of qualitative analysis and the ongoing concerns about generalization might still leave some doubts, but the additional robustness would likely tip the score in favor of acceptance.

---

### Decision · Program_Chairs · 2026-01-26

Accept (Poster)